# Screening the Pathogen Box to Discover and Characterize New Cruzain and *Tbr*CatL Inhibitors

**DOI:** 10.3390/pathogens12020251

**Published:** 2023-02-04

**Authors:** Thales do Valle Moreira, Luan Carvalho Martins, Lucas Abreu Diniz, Talita Cristina Diniz Bernardes, Renata Barbosa de Oliveira, Rafaela Salgado Ferreira

**Affiliations:** 1Molecular Modeling and Drug Design Laboratory, Department of Biochemistry and Immunology, Institute of Biological Sciences, Federal University of Minas Gerais, 6627, Antônio Carlos Avenue, Belo Horizonte 31270-901, MG, Brazil; 2Pharmaceutical Products Department, Federal University of Minas Gerais, 6627, Antônio Carlos Avenue, Belo Horizonte 31270-901, MG, Brazil

**Keywords:** Chagas disease, cruzain, screening, drug discovery, small molecule inhibitors, Pathogen Box

## Abstract

Chagas disease and Human African Trypanosomiasis, caused by *Trypanosoma cruzi* and *T. brucei*, respectively, pose relevant health challenges throughout the world, placing 65 to 70 million people at risk each. Given the limited efficacy and severe side effects associated with current chemotherapy, new drugs are urgently needed for both diseases. Here, we report the screening of the Pathogen Box collection against cruzain and *Tbr*CatL, validated targets for Chagas disease and Human African Trypanosomiasis, respectively. Enzymatic assays were applied to screen 400 compounds, validate hits, determine IC_50_ values and, when possible, mechanisms of inhibition. In this case, 12 initial hits were obtained and ten were prioritized for follow-up. IC_50_ values were obtained for six of them (hit rate = 1.5%) and ranged from 0.46 ± 0.03 to 27 ± 3 µM. MMV687246 was found to be a mixed inhibitor of cruzain (*K_i_* = 57 ± 6 µM) while MMV688179 was found to be a competitive inhibitor of cruzain with a nanomolar potency (*K_i_* = 165 ± 63 nM). A putative binding mode for MMV688179 was obtained by docking. The six hits discovered against cruzain and *Tbr*CatL are of great interest for further optimization by the medicinal chemistry community.

## 1. Introduction

Despite being described over a hundred years ago, Chagas Disease (CD, also American Trypanosomiasis) and Human African Trypanosomiasis (HAT, also sleeping sickness) are still relevant public health problems [1]. While CD accounts for a burden of 546,000 Disability-Adjusted Life Years (DALY) [2] and threatens circa 70 million people with a risk of infection [3], HAT causes a burden of 560,000 DALY [2] and threatens circa 65 million people [4,5]. Moreover, global warming has been shifting CD transmission zones, raising worries in previously transmission-free areas [6], whereas business, commerce, and migration have spread CD prevalence to all inhabited continents except Africa [7]. Addressing the global problem posed by these Neglected Tropical Diseases (NTD) is a costly, long-term investment, as making a novel drug into the market is a billionaire enterprise [8]. Nevertheless, the socio-economic benefits of improving the CD and HAT scenarios are certain [9].

Available treatments are suboptimal for both diseases. Benznidazole and Nifurtimox, the current chemotherapeutic options for CD, are only effective in the acute phase and are associated with significant adverse effects, which include gastric and neurological disorders [3,10,11,12,13]. Moreover, it was shown that chronic asymptomatic patients have no clinical benefit on their cardiac condition after Benznidazole therapy [14]. Likewise, HAT chemotherapeutic resources are highly toxic and low-efficacy drugs [4], which leave many patients untreated, still infected with the parasite. Suramin and Pentamidine are only useful in the early stages of the disease and are associated with severe side effects, including lethal hypoglycemia for Pentamidine [15,16]. Melarsoprol is restricted to the late-stage HAT mainly because 1 out of 10 treated patients will suffer from lethal, reactive encephalopathy [17,18]. Eflornithine is better tolerated than Melarsoprol, but it has to be administered by daily injections [19]. Nifurtimox-eflornithine combination therapy (NECT) presents safety advantages, but still has similar efficacy to eflornithine monotherapy [20,21]. Cost, severe side effects, and complex mode of administration limit Eflornithine mostly to a second-line, late-state treatment [19,22]. Fexinidazole is an oral treatment for late-stage HAT caused by the *T. b. gambiense* strain, with efficacy similar to NECT [23,24,25], and is currently under clinical trials for use against the *T. b. rhodesiense* strain [26]. Therefore, a safe orally administered drug, active against both *T. b. gambiense* and *T. b. rhodesiense,* and either in early or late stages is highly demanded [4,11,12,16].

Cruzain, a cysteine protease of *Trypanosoma cruzi*, is a validated [11,27,28,29] and well-explored pharmacological target (see [30] for a recent review). Cruzain is the truncated, recombinant-expressed cruzipain 1, part of a multigenic family comprising of four subtypes of cruzipains [31]. *Tbr*CatL, a homologous protease from *Trypanosoma brucei,* is also a validated target for discovering trypanocidal compounds [32,33]. Cruzain is expressed throughout the whole life cycle of the parasite [34] and is involved in several vital processes such as replication [35], cellular invasion [36,37], and modulation of macrophage response [38]. *Tbr*CatL might be relevant for the parasite to cross the blood-brain barrier [39], although this subject is still a matter of dispute [40]. Several classes of cruzain inhibitors have been described such as: thiosemicarbazones [41,42], nitrile-based derivatives [43,44], aminoquinolines [45], benzimidazoles [46,47], vinyl sulfones [48,49], analogues of Gallinamide A [50], quinazolines [51], and carbamoyl imidazoles [52]. Likewise, diverse *Tbr*CatL inhibitors have been reported, such as thiosemicarbazones [53], bromoisoxazolines [54], nitriles [55,56], thiazoles and thiazolidines [57,58], triazoles [59], macrocyclic lactams [33], vinyl sulfones [60], vinyl ketones [61], and vinyl esters [62]. Simultaneously investigating the same libraries against these two cysteine proteases can be fruitful due to their similarities, a strategy that has resulted in the description of several inhibitor classes targeting both proteases [11,12,29,42,43,45,60,63,64].

One of the strategies for discovering new lead inhibitors for a target is screening diverse, curated compound sets. For instance, the Medicines for Malaria Venture (MMV) partnership organized the Malaria Box, a compound set for catalyzing NTD research [65]. The Malaria Box was widely screened worldwide, resulting in the discovery of hits against several pathogens and targets [65], including new cruzain and *Tbr*CatL inhibitors [63]. More recently, MMV assembled the Pathogen Box (PB), a compound set targeting a broader scope of parasites (https://www.mmv.org/mmv-open/pathogen-box/about-pathogen-box, accessed on 3 February 2023). PB comprises of 400 drug-like compounds active against parasites that cause NTD and with low-cytotoxic. Some of the molecules originally included in the PB due to in vitro activity against other parasites other than trypanosomatids were later reported to show trypanocidal activity [66,67].

Given the relevance of CD and HAT and the attractiveness of cruzain and *Tbr*CatL as targets, we screened and validated hits from the PB against both enzymes. Out of 400 compounds, we describe six hits with IC_50_ in the low micromolar to nanomolar range against both enzymes, including one competitive inhibitor. These molecules will be very relevant for future medicinal chemistry efforts. 

## 2. Materials and Methods

### 2.1. Data on the PB Collection

PB data spreadsheets containing chemical structures, SMILES, molecular weight, cLogP and biological activity were retrieved from the Pathogen Box website (www.mmv.org, accessed on 30 May 2020). HepG2 cytotoxicity data (CC_20_ and/or CC_50_) are also provided for approximately three-quarters of the compounds. Data on other assays for selected compounds were obtained from the CHEMBL database [68].

### 2.2. PB Collection Samples

The 400 compounds from the Pathogen Box were supplied in 96-well plates containing 10 mM frozen dimethylsulfoxide (DMSO) solutions. Solid samples of the 10 following compounds were resupplied by Evotec upon request to the MMV: MMV688179, MMV688271, MMV667494, MMV634140, MMV690027, MMV690028, MMV688362, MMV085499, MMV687246, MMV688466, MMV687812, MMV688466, and MMV085499.

### 2.3. Assays against Cruzain and TbrCatL

Allison Doak and Prof. Brian Shoichet (University of California San Francisco, San Francisco, CA, USA) and Prof. Conor Caffrey (University of California San Diego, San Diego, CA, USA) generously provided recombinant cruzain and *Tbr*CatL, respectively. In vitro activity of proteases cruzain and *Tbr*CatL was assayed as previously described [45,63]. Briefly, enzyme activity was measured by monitoring the cleavage of the fluorogenic substrate Z-Phe-Arg-amidomethylcoumarin (Z-FR-AMC) at 25 °C. Fluorescence was monitored at 340/440 nm (excitation/emission) over 5–7 min in a Synergy2 Biotek plate reader. Unless stated otherwise, assays were performed using 2.5 μM substrate (*K*_m_ = 0.5 ± 0.1 μM against cruzain and *K*_m_ = 0.5 ± 0.1 μM against *Tbr*CatL [63]) and circa 0.5 nM enzyme in a pH 5.5 buffer composed of 0.1 M sodium acetate buffer, 0.01% Triton X-100, and 1 mM β-mercaptoethanol. DMSO and 1 μM E-64 were employed as negative and positive controls, respectively, in all assays. Two conditions were employed for both enzymes: 10′ pre-incubation of the enzyme in the presence of the compound (10′ inc) and no pre-incubation (0′). Reported values correspond to the mean and standard error of the mean (SEM) and all analyses were performed using GraphPad Prism 6.0.

For the initial screening, compounds were assayed at 5 μM. Compounds were selected for further assays based on the combination of the inhibition observed, novelty, and diversity (see details below). IC_50_ curves were obtained from a non-linear fit of at least seven distinct compound concentrations, in triplicate. Reported IC_50_ values are the mean and SEM of two independent measurements.

To assess the effects of the Triton X-100 concentration on cruzain inhibition, assays were performed using 0.001%, 0.01%, and 0.1% Triton X-100. To evaluate the impact of pre-incubation with bovine serum albumin (BSA), 50 µL of a solution of the compound in the assay buffer containing 0.001% of Triton X-100 was incubated with 4 mg/mL of BSA (Sigma-Aldrich) for 10′ in a 96-well plate. Next, 50 µL of a solution containing circa 2 nM of cruzain in the same buffer was added to each well and incubated for another 10′. Finally, 100 µL of a solution of 5 µM Z-FR-AMC in the same buffer was added to each well and the fluorescence was immediately read. The final compound concentration varies and was selected to be near the compound IC_50_. When analyzing the effects on the inhibition of the concentration of Triton X-100 and pre-incubation of the compound with BSA, differences higher than 20% were considered [51].

The mechanisms of inhibition were determined using seven substrate concentrations (10 µM, 5 µM, 2.5 µM, 1.25 µM, 625 nM, 312.5 nM, 156 nM) and five inhibitor concentrations (including one concentration close to the compound’s IC_50_ value, two concentrations higher and two concentrations lower than the IC_50_) and in the absence of the compound. Non-linear regression to Michaelis-Menten models and linear regression for the Lineweaver-Burk plot were performed. *K_i_* was estimated by both methods. The effect of the inhibitor concentration on the *K*_m app_ was evaluated using the General Linear F-test.

### 2.4. Analogue Search

Analogues of MMV688179 were searched in BraCoLi, a diverse database (containing 1176 molecules previously synthesized by Brazilian medicinal chemistry groups as of October 2022) [69] using DataWarrior. FragFP molecular fingerprints were generated for the entire library and then used to generate a similarity map. The Bemis-Murcko [70] core of MMV688179 was extracted and used as a reference for similarity comparison. Molecules most similar to the core and readily available in our stocks were selected for assays.

### 2.5. Synthesis and Characterization

The synthesis and characterization of compounds 1–5 [71] and 6–7 [72] were previously reported.

### 2.6. Molecular Docking

The crystallographic structure of cruzain was obtained from RSCB PDB (ID: 3KKU [73]) and had hydrogens added using PDB2PQR [74] using a model pH of 5.5. The catalytic dyad was modelled as an ionic pair and Glu208 was modelled in the deprotonated state. MMV688179 was obtained from ChemBL and the most common protomer at pH 5.5, predicted using ChemAxon MarvinSketch (Marvin version 21.15.0, ChemAxon (https://www.chemaxon.com, accessed on 8 December 2022), was the +2 protomer. MMV688179 geometry was optimized at ab initio level using HF/6-31G* in Psi4 [75]. Docking was performed in AutoDock Vina 1.2.3 [76]. The grid was centered at the center of mass of the crystallographic ligand of the protein and exhaustiveness was set to 32. All other parameters for AutoDock Vina were kept at default values. Ten poses were generated and visually inspected for complementarity between the ligand and the binding site, the presence of hydrogen bonds and polar interactions, and the absence of strained torsional angles.

## 3. Results

To assess potential hits against cruzain and *Tbr*CatL, we initially screened the 400 compounds in the PB library at 5 μM against both enzymes (Figure 1). Overall, the dispersion of the results was low as shown by the mean Standard Deviation (SD) of 5.8 percentage points (pp)% over all measurements (N = 1600). Only four compounds (1%: MMV667494, MMV676881, MMV688179, MMV688271) inhibited cruzain or *Tbr*CatL by at least 60%, while 12 compounds (3%: MMV085499, MMV634140, MMV667494, MMV676881, MMV687246, MMV687812, MMV688179, MMV688271, MMV688362, MMV688466, MMV690027, MMV690028) inhibited either enzyme by at least 40% (Figure 2, see also Appendix A). 

Hits were chosen for follow-up based on the combined analysis of their novelty concerning known inhibitors, chemical diversity within the set, and potency in the initial screening. We initially considered for further investigation the 12 compounds that inhibited either cruzain or *Tbr*CatL by at least 40% either without pre-incubation or after 10′ pre-incubation (Figure 3, Table 1). Compound MMV676881 fully inhibited cruzain and *Tbr*CatL both with and without pre-incubation, but it was deprioritized as it is a well-known purine-nitrile cruzain inhibitor [43]. The arylamidine MMV688179 also completely inhibited cruzain both without and with incubation, although inhibition towards *Tbr*CatL was more modest (0’: (50 ± 5)%; 10′ inc: (58 ± 44)%, Table 1). MMV688271, an isomer of MMV688179, inhibited the enzymes to a lesser extent and was not prioritized for follow-up. Similarly, among the structurally related quinolines MMV667494 and MMV634140, the former was prioritized due to its higher potency against *Tbr*CatL after 10′ incubation ((73 ± 1)% vs. (42 ± 3)%). MMV690028 was prioritized because of its *Tbr*CatL inhibition (49 ± 4% without pre-incubation). MMV690027 was deprioritized because of its low cruzain inhibition ((6 ± 2)% without pre-incubation; (0 ± 6)% after 10‘ pre-incubation) and because it was supplied as a racemic mixture. MMV085499 and MMV687246 were prioritized based on the *Tbr*CatL inhibition results after pre-incubation ((46 ± 5)% and (49 ± 4)%, respectively). MMV688179, MMV688271, and MMV688362 have been previously assayed against *T. cruzi* and *T. brucei* and showed IC_50_ between 0.2 µM and 14 µM [77,78] (Appendix A). Surprisingly, the screen also revealed compounds that increased enzyme activity. In the presence of the pyridinylthiazole MMV676409 (Appendix A), we observed higher initial velocities of substrate cleavage, up to 8-fold higher in comparison to the DMSO control (*Tbr*CatL 10′ inc). Compound MMV676512 (Appendix A), which is structurally similar to MMV676409, showed a similar, but less pronounced effect (a 2-fold increase in *Tbr*CatL velocity after 10′ pre-incubation). Thus, we also selected compounds MMV676409 and MMV676512 for further investigation. 

Next, we evaluated the compounds selected from the screening, at a concentration of 100 µM, to confirm their activity using solid samples provided by Evotec (Table 1). Assays for MMV676409 and MMV676512 showed no increase in the enzyme velocity under varying compound concentrations, suggesting that these compounds are not activators of cruzain (Appendix A). Failing in reproducing inhibitions observed in the screen during confirmatory assays is common, so we believe that the lack of activation is not surprising [63]. More specifically, activators have also been observed in an HTS against cruzain but were not confirmed in follow-up assays [79]. MMV085499 is fluorescent at 330/440 nm, thus resulting in interference with the assay and could not be further evaluated. MMV667494 results were not reproducible, as no *Tbr*CatL inhibition was observed in the confirmatory assays. This compound was modestly potent against cruzain after pre-incubation ((70 ± 3)% inhibition), but not potent enough for IC_50_ determination. The remaining six compounds inhibited either or both enzymes to a 90–100% extent, and we determined their IC_50_ against both enzymes with and without pre-incubation (Table 1, see also Appendix A). 

Overall, IC_50_ values determined with and without 10′ pre-incubation were similar (up to 2-fold change for MMV687246 against cruzain), suggesting that none of the compounds is a time-dependent inhibitor. Furthermore, because none of these compounds bears electrophilic moieties, the inhibition likely is a fast-reversible, non-covalent one. Therefore, the IC_50_ and *K*_i_ values can capture the potency of these compounds. In addition, potencies towards cruzain and *Tbr*CatL were similar for all compounds, except for MMV688179, which was the most potent cruzain inhibitor (IC_50_ 0.46 ± 0.03 µM 0’ and 0.53 ± 0.03 10′ inc) but had 10-fold higher IC_50_ against *Tbr*CatL (IC_50_ = 4 ± 1 µM 0’ and 5 ± 2 uM 10′ inc). MMV688362 was the most potent *Tbr*CatL inhibitor (IC_50_ 2.9 ± 0.8 µM 0’ and 2.3 ± 0.1 µM 10′ inc). 

To investigate if inhibition was due to colloidal aggregation, a common cause of unspecific enzyme inhibition in in vitro assays [79], we employed two well-established experiments: the comparison of percentages of inhibition at varying Triton X-100 concentrations and the effect of compound pre-incubation with BSA on the percentage of inhibition. Triton X-100 disrupts small molecule aggregates, so a large reduction in the inhibition observed in high Triton concentration (0.01 or 0.1%) when compared to a low concentration (0.001%) suggests the compound aggregates [80]. Pre-incubating a compound that aggregates with BSA saturates the protein-binding capacity of the aggregate. Therefore, if pre-incubating a compound with BSA reduces the inhibition, it suggests that the compound aggregates [81]. For MMV688362 and MMV687812 we observed some reduction in the inhibition upon the pre-incubation of the compound with BSA and upon a 10-fold increase in the concentration of Triton X-100, respectively (Appendix A). It is worth noting, however, that pre-incubation of MMV687812 with BSA did not reduce the inhibition of cruzain. While these two compounds might possess some aggregation properties near the IC_50_, overall the results do not suggest aggregation. Importantly, results for MMV688179, the most potent cruzain inhibitor, do not suggest aggregation properties near the IC_50_. 

Inhibition mechanisms against cruzain could be determined for MMV688179 and MMV687246 (Figure 4). For MMV687246 a mixed inhibition was observed with an α value slightly smaller than 1 (α = 0.7 ± 0.2) and a *K*_i_ = 57 ± 6 µM (Figure 4A). The lack of substrate competition is also suggested by the Lineweaver-Burk plot (Figure 4B). Accordingly, analysis of the *K_mapp_* at varying concentrations of MMV687246 suggests that the concentration of MMV687246 does not affect the *K*_mapp_ (general linear F-test, *p*-value = 0.1701, Figure 4C). For MMV688179 we clearly observed competition with the substrate, based on the combined analysis of the Michaelis-Menten (Figure 4D) and Lineweaver-Burk plots (Figure 4E). The *K_i_* was estimated to be 165 ± 63 nM from the non-linear regression of the Michaelis-Menten competitive model and 206 nM from the linear regression of *K*_mapp_ (Figure 4F), for which the hypothesis of the slope to be 0 can be rejected (general linear F-test, *p*-value < 0.0001).

Motivated by the high potency and competitive mechanism of MMV688179 against cruzain, we sought to obtain preliminary SAR data on this scaffold. We used a similarity search to find analogues bearing a similar diphenylfurane core in an in-house library (BraCoLi) and obtained seven analogues that were available in our stocks (Figure 5, Appendix A). Unfortunately, we were unable to test most compounds, due to issues with solubility and fluorescence at 330/440 nm. The few compounds tested were much less potent than MMV688179, causing inhibitions of (57 ± 4)% and (65 ± 4)% at 100 µM (Appendix A).

Due to the competitive mechanism and the high potency of MMV688179, we employed molecular docking to propose a binding mode for this molecule. Overall, the ten docking poses were similar and the best-scored one was selected by visual inspection as the possible binding mode of MMV688179. In this binding pose, one of the protonated guanidines is buried in the S2 subsite and an ion-ion interaction with Glu208 is suggested. The other protonated guanidine forms hydrogen bonds with the main chain of residues Gly20 and Cys22, both at the S1 subsite (Figure 6). In addition to these polar interactions, the docking pose spans over a large part of the catalytic cleft, with good spatial complementarity to the protein site, which indicates a high number of van der Waals interactions and is in line with the high potency and competitive mechanism. No interactions were observed for the chlorine atoms, which would rationalize the similar, although slightly lower, inhibition of MMV688271 in the initial screen. In addition, the docking pose suggests the furan ring acts as a linker. However, further studies will be required to shed light on the SAR of MMV688179.

## 4. Discussion

Regarding NTD drug discovery, phenotypic approaches seemed to have been overshadowed by target-based ones over the second half of the 20th century. Their apparent historic contrast has then evolved and merged into complementarity [82]. First-in-class compounds are often discovered in phenotypic approaches whereas further molecular lead optimization follows target-based campaigns. Starting from phenotypical hits, however, may lead to challenges in optimization steps [83]. The MMV chemical boxes are a relevant contribution in that sense, providing phenotypically validated molecules to groups for evaluation in additional phenotypic assays or target-based approaches. The PB compounds are active against a wide range of pathogens but chiefly against *Plasmodium* (125 antimalarials, 33% of the tested molecules), *Mycobacterium* (116, 30%), and kinetoplastids (70, 18%), with the remaining 19% being active against other pathogens [66,77]. Most of the cysteine protease inhibitors discovered in our study have previously shown activity against kinetoplastids. Thus, we contribute toward the deconvolution of the targets related to their trypanocidal activity. For instance, MMV688179 is trypanocidal with an EC_50_ of 27 µM against *T. cruzi* [84], and given the IC_50_ against cruzain we report here, cruzain inhibition is possibly related to its trypanocidal activity. This may also be true for MMV688362 (EC_50_ against *T. cruzi* reported on PB: 13 µM). Both MMV688179 and MMV688362, however, are known to bind to the DNA minor groove, as shown by SPR, ITC, CD, and T_m_ experiments [78,85], which also correlates with the potency against *T. brucei*. Our findings regarding cruzain inhibition suggest, thus, a possible dual mechanism of action. To what extent the trypanocidal effect is derived from the cruzain or DNA binding effect is still to be evaluated by further studies. Finally, it is worth noting that multi-target drugs are an interesting strategy in medicinal chemistry, possibly leading to improved efficacy and overcoming drug resistance [86,87].

The overall hit rate of 1.5% (6 out of 400) observed in our screening was similar to the 1% (4 out of 400) hit rate against cruzain and *Tbr*CatL we reported upon screening MMV’s Malaria Box [63]. PB was also screened against SARS-CoV-2 M^Pro^, a cysteine protease from another family, yielding 2 hits (hit rate = 0.5%). Interestingly, MMV688179 was one of the hits with an IC_50_ of 1.6 µM against M^Pro^ [88]. As MMV688179 does not possess electrophilic moieties (see Figure 3), the M^Pro^ activity is likely due to specific interactions with the enzyme. This is also in line with the lack of MMV688179 inhibition against two unrelated human proteins ferrochelatase and porphobilinogen deaminase [89]. As the reported CC_50_ of MMV688179 is 11.6 µM [89], the selectivity towards the parasite is a clear focus of optimization rounds. It has been reported that cruzain inhibitors can show selectivity towards the parasite [47,51], suggesting that optimizing the cruzain potency of MMV688179 may be an attractive strategy for improving selectivity. Our hit rate is also in line, albeit the much smaller screening library, with the ones reported in HTS campaigns against cruzain (912 out of 197861, 0.46% [73]) and human cathepsin B (20 out of 64000, 0.03% [90]), a highly similar protein. In light of these observations, we believe that screening diverse, curated, yet concise, molecule libraries is an attractive strategy for discovering hits.

It is worth noting that here we report MMV688179 to be a novel, non-covalent cruzain inhibitor bearing a competitive mechanism and nanomolar potency. Unfortunately, we could not determine the potency of a first round of analogues due to interference of the small molecules with the assay readout. Nevertheless, MMV688179 is an interesting candidate for hit-to-lead optimization. This is also true, to a lesser extent, for MMV688362 and MMV687812 which inhibited cruzain and *Tbr*CatL in the 2–4 µM range.

In summary, three main contributions arise from this work. First, our results shed light on possible trypanocidal mechanisms of some of the PB compounds. Second, we add to the evidence that screening diverse and curated, yet small libraries are a useful strategy for discovering new leads. Third, we provide the drug discovery community with novel hits for drug discovery targeting cruzain and *Tbr*CatL. 

## 5. Conclusions

CD and HAT remain relevant, life-threatening diseases affecting mainly disadvantaged populations worldwide. Here, we screened Pathogen Box against cruzain and *Tbr*CatL, two validated targets for discovering leads against *T. cruzi* and *T. brucei*, respectively. From the 400 compounds in the library, we obtained 12 hits and validated six of them as inhibitors of cruzain and *Tbr*CatL. Particularly, MMV687246 is a mixed inhibitor of cruzain with a *K_i_* of 57 ± 6 µM and MMV688179 is a competitive inhibitor of cruzain with a *K_i_* of 165 ± 63 nM. We also proposed a possible binding mode of MMV688179 to cruzain to aid further optimization efforts. Hit-to-lead optimization of MMV688179 should focus on increasing selectivity towards *T. cruzi* and improving its potency against cruzain may be an attractive strategy in that context. We believe that the molecules discovered in this work, especially MMV688179 are candidates for optimization. 

## Figures and Tables

**Figure 1 pathogens-12-00251-f001:**
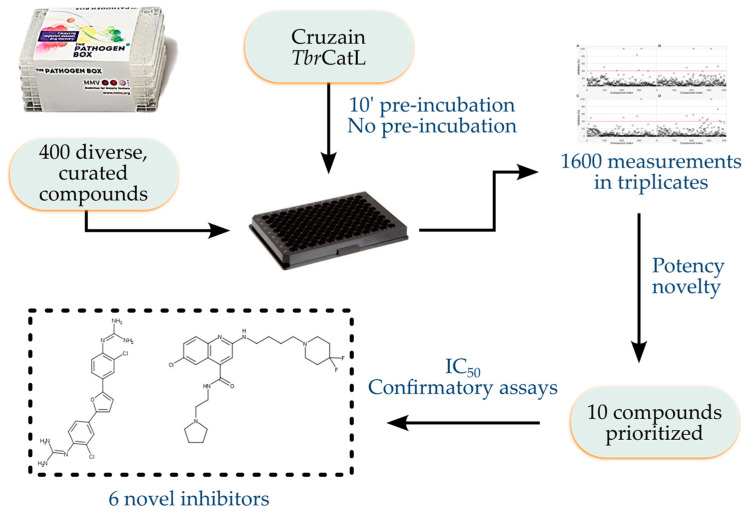
Workflow of the study. The 400 compounds of the Pathogen Box set were assayed for inhibition of cruzain and *Tbr*CatL. The 1600 inhibition measurements were analyzed and compounds inhibiting >40% of the enzyme activity against either cruzain or *Tbr*CatL were considered for prioritization. Ten compounds were prioritized and six of them were confirmed to be inhibitors with IC_50_ ranging from 0.46 ± 0.03 to 27 ± 3 µM.

**Figure 2 pathogens-12-00251-f002:**
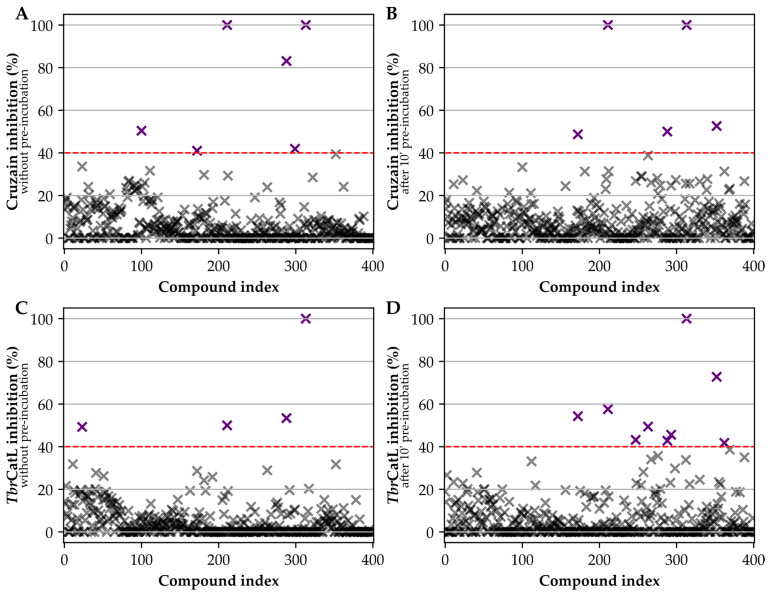
Scatter plots of mean inhibitions observed in the screening at 5 µM. (**A**) Inhibition of cruzain without pre-incubation. (**B**) Inhibition of cruzain after 10′ pre-incubation. (**C**) Inhibition of *Tbr*CatL without pre-incubation. (**D**) Inhibition of *Tbr*CatL after 10′ pre-incubation. Values below 40% are depicted as semi-transparent grey crosses. Values above 40% are depicted as purple crosses.

**Figure 3 pathogens-12-00251-f003:**
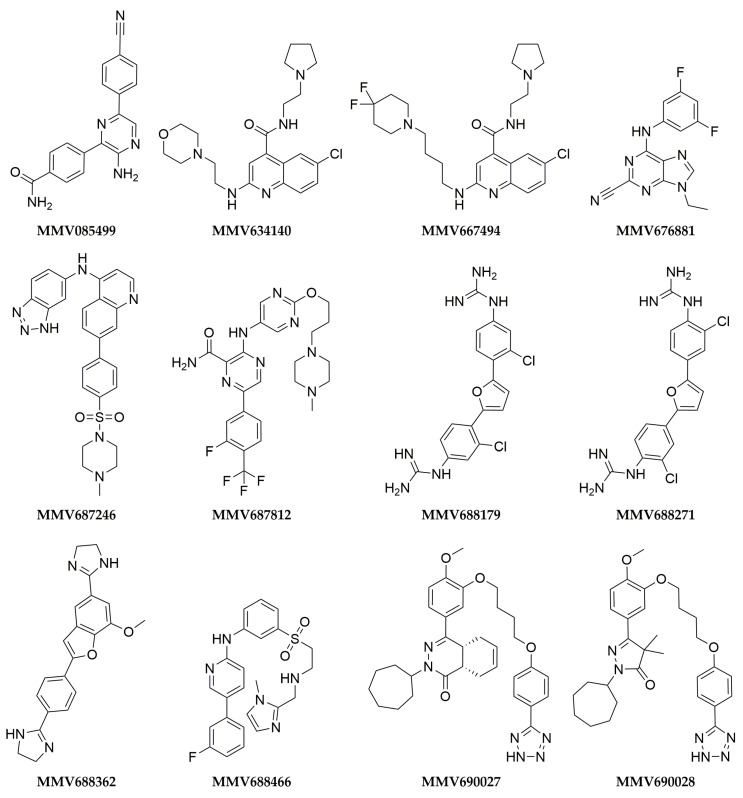
Structures of compounds inhibiting cruzain or *Tbr*CatL by at least 40% in the initial screening at 5 µM.

**Figure 4 pathogens-12-00251-f004:**
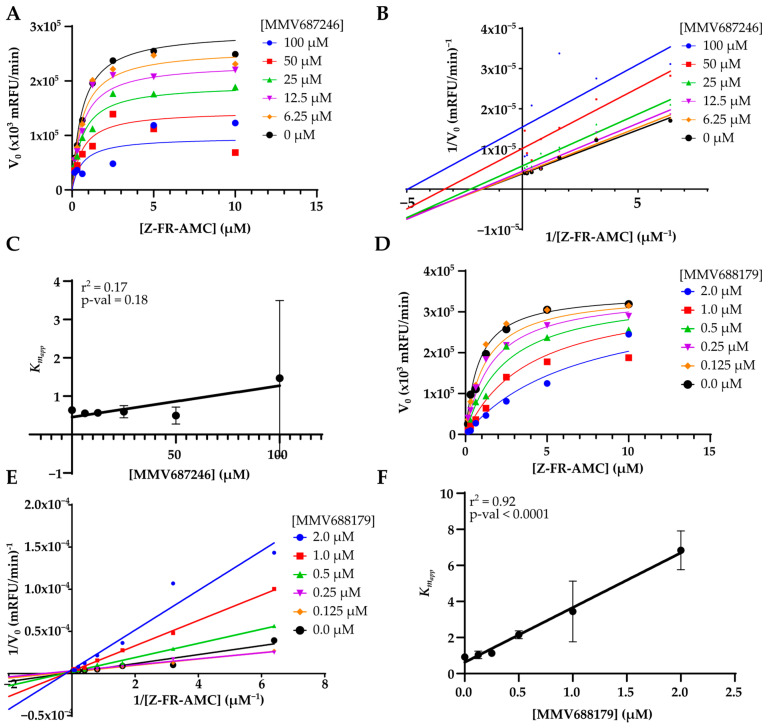
Plot of the assays to determine the mechanism of inhibition of MMV687246 and MM688179. (**A**) Michaelis-Menten plot for MMV687246. The curves correspond to the fitting of a mixed-model of inhibition to data. (**B**) Lineweaver-Burk plot for MMV687246. (**C**) Plot of the *K*_mapp_ for varying MMV687246 concentrations. Points are the mean of two replicas. *p*-val is the *p*-value of the general linear F-test. (**D**) Michaelis-Menten plot for MMV688179. The curves correspond to the fitting of a mixed-model of inhibition to data. (**E**) Lineweaver-Burk plot for MMV688179. (**F**) Plot of the *K*_mapp_ for varying MMV688179 concentrations. Points are the mean of two replicas. *p*-val is the *p*-value of the general linear F-test.

**Figure 5 pathogens-12-00251-f005:**
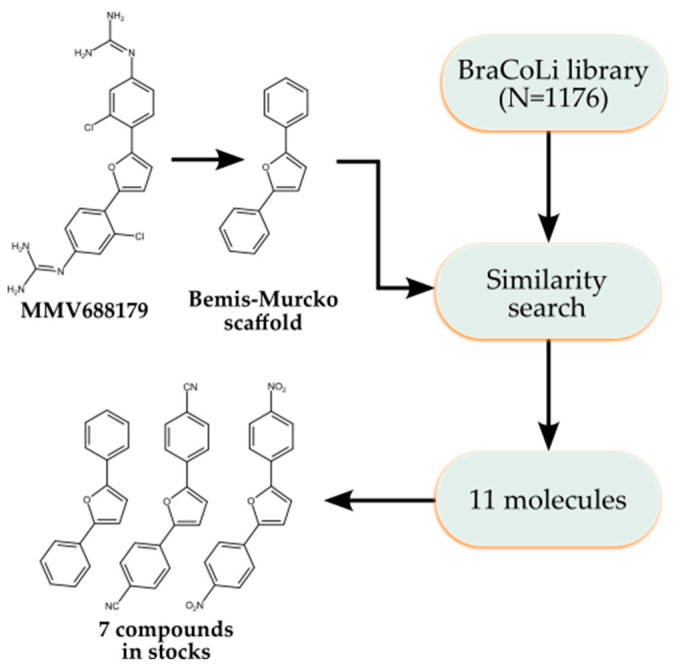
Workflow for searching for MMV688179 analogues. The Bemis-Murcko scaffold of MMV688179 was calculated and used as a reference for the similarity search of analogues in the BraCoLi dataset. Here, 11 molecules were selected and seven of them were in stock.

**Figure 6 pathogens-12-00251-f006:**
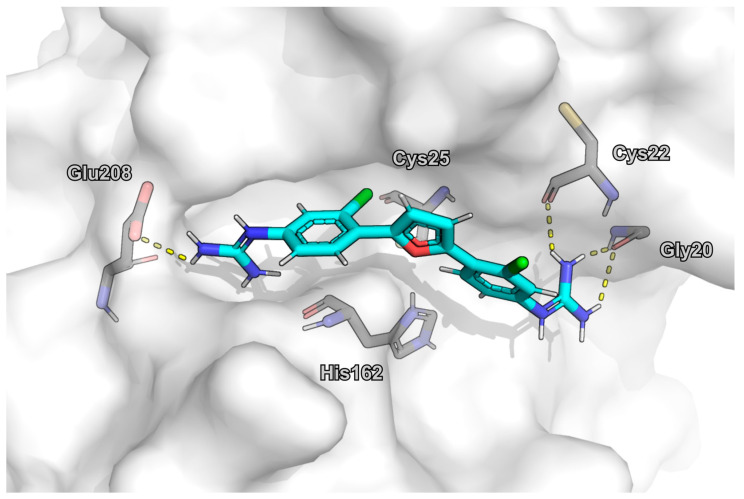
Docking pose of MMV688179 bound to cruzain’s active site. Figure prepared in PyMol 2.4.0.

**Table 1 pathogens-12-00251-t001:** *Tbr*CatL and cruzain inhibition by hits prioritized after the initial screening of the Pathogen Box.

Compound	Inhibition at 5 µM (Mean ± SEM %) ^a^				Inhibition at 100 µM (Mean ± SEM %) ^b^				IC_50_ (µM ± SEM %) ^c^			
	*Tbr*CatL		Cruzain		*Tbr*CatL		Cruzain		*Tbr*CatL		Cruzain	
	0′	10′ inc	0′	10′ inc	0′	10′ inc	0′	10′ inc	0′	10′ inc	0′	10′ inc
MMV085499	12 ± 8	46 ± 5	2 ± 3	25 ± 4	26 ± 18	0 ± 0	0 ± 0	0 ± 0	ND	ND	ND	ND
MMV634140	11 ± 1	42 ± 3	24 ± 1	31 ± 3	ND	ND	ND	ND	ND	ND	ND	ND
MMV667494	31 ± 5	73 ± 1	40 ± 1	53 ± 1	0 ± 0	11 ± 8	0 ± 0	70 ± 3	ND	ND	ND	ND
MMV676881	100 ± 0	100 ± 2	100 ± 2	100 ± 1	ND	ND	ND	ND	ND	ND	ND	ND
MMV687246	29 ± 1	49 ± 4	24 ± 3	39 ± 3	92 ± 6	99 ± 1	72 ± 1	92 ± 3	9 ± 3	4.2 ± 0.6	14 ± 5	10 ± 2
MMV687812	11 ± 3	5 ± 1	41 ± 2	27 ± 0	90 ± 2	93 ± 0	100 ± 0	95 ± 0	3.45 ± 0.05	3.6 ± 0.3	2.2 ± 0.3	2.6 ± 0.03
MMV688179	50 ± 3	58 ± 25	98 ± 2	100 ± 0	100 ± 0	100 ± 0	100 ± 0	100 ± 0	4 ± 1	5 ± 2	0.46 ± 0.03	0.53 ± 0.03
MMV688271	53 ± 5	43 ± 1	83 ± 10	50 ± 4	ND	ND	ND	ND	ND	ND	ND	ND
MMV688362	9 ± 1	43 ± 4	19 ± 1	27 ± 3	100 ± 0	100 ± 0	100 ± 0	100 ± 0	2.9 ± 0.8	2.3 ± 0.1	4.2 ± 0.6	2.35 ± 0.02
MMV688466	29 ± 2	54 ± 8	41 ± 2	49 ± 5	96 ± 3	100 ± 0	58 ± 27	98 ± 1	4 ± 1	11 ± 4	9 ± 1	25 ± 6
MMV690027	6 ± 2	0 ± 6	50 ± 5	33 ± 2	ND	ND	ND	ND	ND	ND	ND	ND
MMV690028	49 ± 4	23 ± 4	34 ± 8	27 ± 4	100 ± 0	88 ± 3	100 ± 0	100 ± 0	18 ± 7	27 ± 3	10 ± 3	13 ± 5

ND = not determined ^a^: mean over three independent measurements, values larger than 100% were represented as 100% and values lower than 0% were represented as 0%. ^b^: mean over two independent experiments, in triplicates. ^c^: mean over two independent experiments, in triplicates, spanning at least seven compound concentrations.

## Data Availability

Data on Pathogen Box compounds was obtained from the MMV website (www.mmv.org, accessed on 30 May 2020) and ChEMBL (www.ebi.ac.uk/chembl, accessed on 25 November 2022). The BraCoLi library is publicly available at https://www.farmacia.ufmg.br/qf/downloads, accessed on 12 August 2022. All other data presented in this study are available within the paper and Appendix A.

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
