# Peer review of "Screening the Pathogen Box to Discover and Characterize New Cruzain and TbrCatL Inhibitors"

_pathogens, 2023, doi:10.3390/pathogens12020251_

Round 1

Reviewer 1 Report

The article in question reports the screening of the Pathogen Box (PB, 400 compounds) against cruzain and TbrCatL of Trypanosoma cruzi and T. brucei, respectively. Twelve hits were identified and six of them were validated as inhibitors of cruzain and TbrCatL. Mechanism of action and binding mode were proposed.

Generally, the article is well-written and properly detailed.

Major points:

However, I would like to highlight important criticism.

In the text, line 344 “In summary, three main contributions arise from this work. First, our results shed light on possible trypanocidal mechanisms of some of the PB compounds. Second, we add to the evidence that screening diverse and curated, yet small libraries is a useful strategy for discovering new leads. Third, we provide the drug discovery community with high-potency hits for structure-based drug discovery against T. brucei and T. cruzi.”

I partially agree with the first and second contributions, but I disagree with the third one.

The most promising compound (MMV688179) showed a Ki value of 332 nM against cruzain, and several inhibitors with similar or better activities are reported in the literature.

Since MMV688179 are a well-known DNA minor groove binder, it should be considered a dual inhibitor.

An important discrepancy between the inhibitory properties towards cruzain (Ki in the double-digit nanomolar range), and the trypanocidal activity of MMV688179 (T. cruzi, EC50 = 27 µM) can be observed. In light of this, could the trypanocidal activity of MMV688179 mainly assess to DNA interaction?

Several important issues (solubility, interferences) during the biological evaluation were found.

Docking studies are not so detailed. Only two interactions were observed. The two residues of Arg can interact with a lot of residues. Is furan ring important? Are the two chlorine atoms important? I wonder if the only two interactions of Arg residues can lead to a Ki in the nanomolar range. How an inhibitor with only two interactions can be considered a lead compound?

MMV688179 was also tested against Mpro of SARS-CoV-2 and an interesting IC50 value was observed. Therefore, can this molecule indiscriminately react with a large panel of cysteine protease? Could human cysteine proteases (hCatL, hCatB, hCatS, etc.) be sensitive to MMV688179?

Overall, could MMV688179 consider a high-potency lead compound for future structure-based optimization of cruzain and TbrCatL inhibitors?

My answer is no.

In light of what I reported above, MMV688179 should be considered a competitive inhibitor of cruzain, with an interesting Ki value, dual mechanism of action (important!), and trypanocidal activity in the micromolar range (already known), and nothing else.

I strongly suggest to modify the article limiting to report the screening, and the identification and biological characterization of the inhibitors. The findings reported in this paper are important, but not so crucial.

Additionally, find below the minor points:

Several times, the authors associate Ki values with potency. Ki indicates the dissociation constant describing the binding affinity between inhibitors and the enzyme. k2nd indicates potency for irreversible inhibitors. So, why do the authors report potency with Ki value? This is misleading. Please, clarify.

Line 22 = “bearing” should be replaced (no appropriate term in that context)

Line 33 = “They account for burdens of 546,000 and 560,000 Disability-Adjusted Life Years [2], and threaten circa 70 and 65 million people with risk of infection [3–5], respectively.” Respectively for what? Both diseases are described in the previous sentence. Please, reformulate.

Line 36 = replace “but” with “except”

Line 45 = which leaves. The sentence is in the plural (resources), so “which leave” is the correct form. Please, replace.

Line 51 = please mention NECT as the acronym

Line 53 = add a comma after “effects”

Line 55 = Please, have a look between “with” and “efficacy”

Line 71 = to the best of my knowledge, 3-bromoisoxazoline derivatives inhibit a panel of cysteine proteases, but not cruzain. Ref 46 describes 3-bromoisoxazoline analogues towards TbrCatL. Please revise or clarify.

Line 75 = Michael acceptors with potent affinity and potency towards TbrCatL were omitted. Please cite 1) 10.1074/jbc.M109.014340 and 2) 10.1021/acs.jmedchem.9b00908

Line 86 = revise this sentence. Is “potent” needed?

Line 92 = the same as line 86. Do you mean “six potent hits”? Please, clarify.

Line 132 = add the full name of BSA, please. 50 µL of a solution of the compound. What is the concentration? 5 µM? Please clarify

Line 177 and 178 = to make the life of the readers easier, list the four and twelve compounds, please.

Line 185 = toward which enzyme? Please clarify.

Line 204 = 42±3% should be 42 ± 3%

Line 224 = MMV compounds in solutions increased the enzyme velocity, while the same effect was not observed for fresh solutions of the same compounds. How the authors can justify that?

Line 243 = could the word “but” be removed? Please, have a look.

From line 249 to line 259 = Since the colloidal aggregation is a key point in the biological assessment, this part should be clearly described. I suggest to properly revise this part in terms of description because I found it very confusing.

Line 262 = target is missing. Please, add it.

Line 325 = the development of multitarget ligands is a valuable strategy in the field of medicinal chemistry. This point should be emphasised (see 1) 10.2174/0929867328666210810125309 and 2) 10.1039/C4MD00069B). 

Reviewer 2 Report

Dear authors,

Please find herein the comments about the manuscript pathogens-2146430.

- Introduction line 57-60 “Although oral administration might lead to increased access to  treatment in remote areas, severe side effects could impair treatment effectiveness [27]. Therefore, a safe orally administered drug, active against both T. b. gambiense and T. b. rhodesiense, and either in early or late stages is highly demanded [4,11,12,16].” This paragraph needs to be reworked. In fact, in the article 27 cited in reference and in the clinical trials of fexinidazole, the undesirable effects reported are not serious. The main ones are digestive disorders. The majority of adverse events were mild or moderate and none led to discontinuation treatment. The use of fexinidazole is not limited by these side effects.

- Line 143 “…higher and two concentrations lower than the IC50) and in its absence.” It was not clear. It would be necessary to specify the absence of what, of the inhibitor ?

- Line 192 to 215: Results not appearing in the table 1 are mentioned. It would be preferable to also show the results of the less active molecules in Table 1, since they are discussed in the text. For example molecules MMV676881, MMV688271...

- Line 239 to 245 : it is written “MMV688179, which was the most potent cruzain inhibitor (IC50 = 0.45 ± 0 µM 0' and 0.52 ± 0 10' inc) but had 10-fold higher IC50 against TbrCatL (IC50 = 4.0 ± 0.8 µM 0' and 4.7 ± 1.6 uM 10' inc)”. But in table 1 we understand that this compound is the most potent inhibitor against TbrCatL : IC50 of compound MMV688179 is 4 µM 0’ and  5 µM 10’ for cruzain, and 0,46 µM 0’ and 0,53 µM 10’ for TbrCatL. Similarly, the results in Table 1 indicate that compound MMV688362 is the most potent cruzain inhibitor (IC50 = 2.9 0' and 2.3 10' and not the most potent tbrCatL inhibitor (IC50 = 4,2 0’ and 2,35- 10’). The concordance of the results between the text and the table must be checked.

- it would be interesting for the six compounds inhibited either or both enzymes to a 90-100%, to specify the cytotoxicity and the trypanocidal activity in table 1.

Round 2

Reviewer 1 Report

Dear Editor,

The quality of the manuscript entitled “Screening the Pathogen box to discover and characterize new cruzain and TbrCatL inhibitors” by Rafaela Salgado Ferreira and co-workers was significantly improved after the revision.

I suggest to accept it for publication after the following minor revisions:

- Table S1. Add the unit of measurement in the cytotoxicity column. For MMV688179, “b” should be superscript in the “IC50 against T. brucei rhodesiense (μM)” column.

- If the CC50 values were reported in µM, the most promising compound (MMV688179) showed CC50 of 11.6 µM and antitrypanosomal activity against T. cruzi of 27 µM. In light of this, I strongly suggest to report in the conclusion that future efforts/optimization should be focused to improve the selectivity towards protozoa.

- Please revise how to report the cited papers in the list of references. In some of them the journal name is abbreviated, in others it is fully reported (for instance, see ref 33 and 43). Please follow the guidelines of the journal.
